

# Antibiotic resistance potential of the healthy preterm infant gut microbiome

Graham Rose[1], Alexander G. Shaw[2], Kathleen Sim[2], David J. Wooldridge[1], Ming-Shi Li[2], Saheer Gharbia[1], Raju Misra[1] and John Simon Kroll[3]

[1] Genomics Research Unit, Public Health England, London, United Kingdom
[2] Department of Medicine, Imperial College London, London, United Kingdom
[3] Section of Paediatrics, Department of Medicine, Imperial College London, London, United Kingdom

## ABSTRACT

**Background.** Few studies have investigated the gut microbiome of infants, fewer still preterm infants. In this study we sought to quantify and interrogate the resistome within a cohort of premature infants using shotgun metagenomic sequencing. We describe the gut microbiomes from preterm but healthy infants, characterising the taxonomic diversity identified and frequency of antibiotic resistance genes detected.

**Results.** Dominant clinically important species identified within the microbiomes included *C. perfringens*, *K. pneumoniae* and members of the *Staphylococci* and *Enterobacter* genera. Screening at the gene level we identified an average of 13 antimicrobial resistance genes per preterm infant, ranging across eight different antibiotic classes, including aminoglycosides and fluoroquinolones. Some antibiotic resistance genes were associated with clinically relevant bacteria, including the identification of *mecA* and high levels of *Staphylococci* within some infants. We were able to demonstrate that in a third of the infants the *S. aureus* identified was unrelated using MLST or metagenome assembly, but low abundance prevented such analysis within the remaining samples.

**Conclusions.** We found that the healthy preterm infant gut microbiomes in this study harboured a significant diversity of antibiotic resistance genes. This broad picture of resistances and the wider taxonomic diversity identified raises further caution to the use of antibiotics without consideration of the resident microbial communities.

Corresponding author
Graham Rose, graham.rose@phe.gov.uk

## INTRODUCTION

Over recent years the composition of the gastrointestinal (GI) microbiota has been increasingly implicated in health and disease, with bacterial populations harbouring both beneficial commensals and pathogens. A diverse bacterial population results in greater genetic content, but some of this additional genetic material is less welcome. Previous studies have implicated the GI microbiota as a reservoir of antimicrobial resistance (AMR) genes (*Penders et al., 2013*), held by, or capable of being transferred to, potential pathogens. Whilst often benign, during bacterial infection transfer of AMR genes can occur, which—coupled with selection pressures arising through antimicrobial therapy—can make treatment difficult, increasing the time taken to cure the infection. Furthermore, antimicrobial therapies are generally (ideally) tailored towards acute infections targeting

a single pathogen, with little consideration of the wider microbial communities which reside in the microbiome, leading to a situation in which the use of antibiotics may cause unintentional harm to the host.

As our understanding of the microbiome has developed, the collection of AMR genes within a bacterial population has recently been defined as the resistome (*Penders et al., 2013*). Antibiotics have a role in shifting the profile of the resistome within the population (*Jernberg et al., 2007*), with low antibiotic-use communities harbouring lower AMR gene frequencies (*Walson et al., 2001*; *Bartoloni et al., 2009*). Heavy treatment of bacterial populations with antibiotics can lead to the long term overrepresentation of AMR genes. Such dynamics are evident in the microbiome of preterm neonates, who receive multiple antibiotic courses, and are cared for in an Intensive Care Unit environment potentially contaminated with multi-resistant bacteria. Antibiotic treatments for both term and preterm neonates have demonstrated lasting effects on the microbiota (*Tanaka et al., 2009*; *Arboleya et al., 2015*), with the trajectory of population development diverging from untreated controls, leading to a potential scenario of prolonged—even life-long—high frequency AMR reservoirs through the selection of bacteria within the population that are most resistant. A wide range of AMR genes have been found in neonatal populations (*De Vries et al., 2011*; *Zhang et al., 2011*), some shown to be present from birth (*Alicea-Serrano et al., 2013*; *Gosalbes et al., 2016*), whilst twin pairs have been shown to have GI communities with similar distributions of both organisms and resistance genes (*Moore et al., 2015*). These observations suggest vertical transmission as a source, with discrepancies between mothers and babies being due to the substantial shifts in the microbiota adapting to the very different environment of a newly born infant's GI tract (*Gosalbes et al., 2016*).

The GI tract of a premature neonate is a particularly unusual scenario for observation of AMR genes, due to greatly reduced bacterial immigration as a result of the isolated, sterile environment of incubators and very controlled enteral feeds; donor breast milk may be pasteurised and, whilst unpasteurised maternal milk (which harbours specific bacteria (*Beasley & Saris, 2004*; *Jimenez et al., 2008*; *Martin et al., 2009*)) is given where possible, there is a likelihood of little or no breastfeeding due to extreme prematurity.

In these circumstances, the GI community and the resistance genes present are likely in the main to be derived from the mother, and acquired during birth. Whilst limited bacterial numbers and diversity will initially be transferred, mechanisms are available for the dissemination of AMR through the expanding bacterial population (as reviewed by *Van Hoek et al. (2011)*) with transfer having been documented within the gut environment (*Shoemaker et al., 2001*; *Karami et al., 2007*; *Trobos et al., 2009*). Heavy use of antibiotics in the course of care of premature infants would not only then skew the bacterial population and drive resistance selection, but has been shown to increase the activity of some transposable elements due to stressing of bacterial populations (*Beaber, Hochhut & Waldor, 2004*).

In this study, we present a detailed investigation of the resistome from the GI microbiota of eleven premature infants, with detailed information on antibiotic receipt and maternal antibiotic use. The microbiota of premature infants has been subjected to such investigations before, but through targeted techniques such as PCR or qPCR

(*Gueimonde, Salminen & Isolauri, 2006*; *Alicea-Serrano et al., 2013*; *Von Wintersdorff et al., 2016*) or through functional metagenomics (*De Vries et al., 2011*; *Moore et al., 2015*), which has the disadvantage of not being able to quantify the antibiotic resistance potential of a community (*Forslund et al., 2014*). We have used shotgun metagenomic sequencing to describe the resistome in its entirety, moving from species level taxonomic profiling, to characterisation of the resistance landscape, including typing of metagenomes identified as potentially harbouring *mecA*, conferring resistance to methicillin and other $\beta$-lactam antibiotics.

## MATERIALS & METHODS

### Study population

The study was approved by West London Research Ethics Committee (REC) Two, United Kingdom, under the REC approval reference number 10/H0711/39. Parents gave written informed consent for their infant to participate in the study.

Faecal samples analysed were collected from premature infants, defined as less than 32 completed weeks of gestation. Premature infants were recruited to the study at the Imperial College Healthcare National Health Service Trust neonatal intensive care unit (NICU), at Queen Charlotte's and Chelsea Hospital, between January 2010 and December 2011.

### Sample collection

Almost every faecal sample produced by each participant between recruitment and discharge was collected by nursing staff from diapers using a sterile spatula. Samples were placed in a sterile DNase-, RNase-free Eppendorf tube, stored at −20 °C within two hours of collection and stored at −80 °C within five days. A single faecal sample from each of twelve infants who had no diagnosis of necrotising enterocolitis or blood-stream infection during their admission was selected for metagenomic sequencing. DNA from one faecal sample did not complete library preparation (see below); clinical characteristics of the remaining eleven infants and faecal sample metadata are presented in Table S1.

### DNA extraction and shotgun library preparation

DNA extractions were performed as described previously (*Rose et al., 2015*), but with the following modifications: DNA extracts were prepared from approximately 200 mg of faeces, which were re-suspended in 10× volume:weight filtered 1× phosphate-buffered saline (PBS), with addition of 1:1 (volume:volume) 2% 2-mercaptoethanol diluted in 1× filtered PBS. The MolYsis selective lysis kit (Molzym) was used for the selective lysis of eukaryotic cells, incorporating the modifications previously described (*Rose et al., 2015*). Bacterial lysis was performed by addition of 50 μl lysozyme (Sigma), 6 μl mutanolysin (Sigma) and 3 μl lysostaphin (Sigma) to 100 μl of re-suspended bacterial pellet, and incubated at 37 °C for 1 h. This was followed by addition of 2 μl proteinase K and 150 μl 2× Tissue and Cell lysis buffer (Epicentre) and incubated at 65 °C for 30 min. Lysates were added to 2 ml tubes containing 0.25 ml of 0.5 mm beads and beaten on a Fast Prep 24 system at 6 m/s for 20 s and repeated once after 5 min. Finally, DNA was purified using the MasterPure complete kit (Epicentre) according to the manufacturer's instructions, eluted in 50 ul 0.1× TE buffer (Sigma) and stored at −80 °C.

Extracted DNA was fragmented using the NEBNext dsDNA fragmentase kit (NEB) according to the manufacturer's instructions. Shotgun DNA libraries were subsequently prepared using the KAPA HyperPrep kit (KAPA Biosystems) according to the manufacturer's instructions. Ligated libraries were amplified by PCR with the number of cycles being dependant on starting material biomass, varying between two and eight (mean three cycles). A negative extraction control was included consisting of 1 ml filtered 1× PBS and processed alongside the samples. After library amplification, the negative extraction control and one preterm infant faecal sample required >8 PCR cycles owing to very low starting pre-PCR biomass (DNA concentration <0.1 ng/ul), therefore these samples were excluded from downstream analysis, leaving faecal samples from eleven premature infants.

## Shotgun metagenomic sequencing

Library insert size and quantity was assessed for each sample by Bioanalyser and qPCR as described previously (*Rose et al., 2015*). Library insert size ranged from 244 bp to 288 bp with a mean of 261 bp. Libraries were sequenced on either an Illumina NextSeq 500 system or part of replicate runs on an Illumina MiSeq system. Prior to loading, libraries were normalised, pooled and diluted to either 1.6 pM or 18 pM for sequencing on the NextSeq or on the MiSeq, respectively. Paired end (PE) sequencing was performed on the NextSeq using a v2 300 cycle high output reagent kit (Illumina) and on the MiSeq using the v3 600 cycle reagent kit (Illumina).

Prior to sequencing the complete dataset, three libraries from three infant faecal samples were sequenced as part of separate multiplexed MiSeq runs, generating a mean 8.3 million PE reads and 5.0 Gbp sequence yield per sample. Later sequencing on a NextSeq instrument included the complete 11 sample set, and inclusion of a technical replicate (sample Q89). A mean 10.0 million PE reads were generated per sample, yielding 3.0 Gbp.

## Sequencing data availability

All sequencing data generated for this study is available from the EBI European Nucleotide Archive, under study accession PRJEB15257.

## Processing of metagenomic sequences

Sequence quality was calculated using FastQC (v0.11.3) (*Andrews, 2010*). Read filtering was performed using Trimmomatic (v0.32) (*Bolger, Lohse & Usadel, 2014*). This consisted of adapter sequence removal based on an in house database of Illumina adapters, primers and index sequences using non-default parameters (*ILLUMINACLIP 2:30:10*), and subsequent read trimming consisting of an initial head crop of the first 15 bp, then iterative removal of leading and trailing bp with phred qualities <20, and internal bases where mean base phred qualities <20 in 4 bp sliding windows (parameters were: *LEADING:20 TRAILING:20 SLIDINGWINDOW:4:20*). Finally sequences with less than 40 bp remaining were discarded (*MINLEN:40*).

A mapping based approach was used to remove expected human host sequences from the faecal samples, as well as any remaining vector contamination. Using FastQ Screen (v0.4.4) (Babraham Institute) and the short read aligner Bowtie2 (v2.2.6) (*Langmead &*

*Salzberg, 2012*), reads were mapped against the human genome (GRCh38) and the UniVec (version 8) vector database (non-default parameters were: *–aligner bowtie2*, *–nohits*). All unmapped PE reads were output as new fastq files and continued within downstream analysis.

The MiSeq replicate datasets entered an identical workflow, except for an additional step designed to utilise the longer read lengths by joining the read pairs using FLASH (v1.2.11) (*Magoč & Salzberg, 2011*) (*–max-overlap 200*), thus generating longer single sequences (mean length 207 bp).

## Species identification and relative abundances

Primary metagenomic profiling was performed using DIAMOND (v.0.7.9.8) (*Buchfink, Xie & Huson, 2014*) and MEGAN (v5.10.6) (*Huson et al., 2007*). All forward reads, or joined reads in the case of the MiSeq dataset, were aligned against a protein reference database under default parameters. The protein database was built using all 73,055,898 sequences from the NCBI non-redundant (nr) database (downloaded 21-10-15).

Processing and taxonomic analysis of the sequence reads with matches to the nr database was performed within MEGAN and under non-default parameters. Sequences were assigned to the NCBI Taxonomy (1,266,115 individual taxa) using the Lowest Common Ancestor (LCA) and the following thresholds: minimum bit-score: 80; max expectation value: $1.0 \times 10^{-6}$; top percentage of hits considered: 10%; minimum taxon support based on all assigned reads: 0.01%. Relative abundances and extraction of species specific binned reads were calculated within MEGAN.

A secondary taxonomic profiling method, MetaPhlAn (v.2.2.0) (*Truong et al., 2015*) was used on all samples with the following parameters: *–mpa_pkl metaphlan2/db_ v20/mpa_v20_m200.pkl –bowtie2db metaphlan2/db_v20/mpa_v20_m200 –input_type fastq*. Relative abundance tables were combined using the packaged MetaPhlAn script— merge_metaphlan_tables.py.

## Identifying antimicrobial resistance genes

Presence/absence testing of AMR genes within the samples was performed on unassembled reads. Reads were mapped using bowtie2 (v.2.2.6) (*Langmead & Salzberg, 2012*) as part of SRST2 (v0.1.7) (*Inouye et al., 2014*) under default parameters to a clustered ARG-Annot database of acquired resistance genes (*Gupta et al., 2014*) and all hits recorded (*–gene_db srst2/data/ARGannot.r1.fasta*). Default parameters set AMR gene reporting at 90% minimum coverage cutoff. Computational *S. aureus* sequence typing (ST) was also performed using SRST2 (*Inouye et al., 2014*) under default scoring parameters, and using the *S. aureus* MLST schema downloaded on 18-04-16 from pubmlst.org. Alleles were flagged uncertain when below threshold depths (–min_edge_depth 2, –min_depth 5).

## Metagenome assembly and *S. aureus* phylogeny

Assemblies were performed using spades (v3.7.1) under default parameters except identification of the data as metagenomic (–meta). Assembled contigs were used as blastn queries against the NBCI nt database, and taxonomic labels attached using MEGAN, with all contigs identified as *S. aureus* (NCBI taxon id: 1280) including summarised contigs

extracted per sample. As a reference, all *S. aureus* complete genomes were downloaded from PATRIC (Release May 2016), totalling 118 genomes.

An anchor based phylogenetic method, andi (v.0.10) (*Haubold, Klötzl & Pfaffelhuber, 2015*), was used to estimate the evolutionary distances between the study and global *S. aureus* genomes set, using PHYLIP (v.3.696) (*Felsenstein, 1989*) to infer the neighbour-joining phylogeny. Following assembly, very short binned *S. aureus* contigs (<1 kb) and partial assemblies, in this case those with less than half of the median *S. aureus* genome size (<1.5 Mb), were excluded from andi and phylogenetic tree construction as based on recommended guidelines (*Haubold, Klötzl & Pfaffelhuber, 2015*).

### *S. aureus* typing

Experimental confirmation of *mecA* was attempted for all eleven samples. Faecal samples were cultivated on the *Staphylococcal* selective growth media manitol salt agar. Sweeps of the presumptive *Staphylococcus* colonies from each sample were propagated and extracted by the following protocol: half a 10 µl loop of overnight growth at 35 °C was inoculated into 2 ml tubes containing 0.5 mm silica/zirconia beads filled to the 0.25 ml mark and 350 µl of Master Pure Tissue Cell Lysis Buffer (EpiCentre). Bead beating was performed using a Fast Prep (MPBio) at 6 m/s for 20 s. This was repeated three times with a 5 min pause between each pulse. Lysates were centrifuged at 8,000× *g* for 10 min and 300 µl of supernatant transferred to a new tube. A known *mecA* positive strain (NCTC strain 12232) and a *mecA* negative clinical isolate were used as control strains.

A multiplexed PCR method was used to type the SCCmec element within the samples according to the protocol described previously (*Milheiriço, Oliveira & De Lencastre, 2007*), but with the following exceptions. Each 50 µl PCR reaction consisted of 1× HotStart Ready Mix (KAPA Biosystems), 25 ng genomic DNA, and primers at the described concentration. The cycling conditions were as follows: 95 °C for 3 min followed by 30 cycles of 98 °C for 20 s, 53 °C for 30 s and 72 °C for 30 s followed by a final extension of 72 °C for 4 min. Amplicons were purified using the AgenCourt AMpure XP PCR purification kit (Beckman Coulter) following manufacturer's instructions. Amplicon sizes were determined measured by BioAnalyser (Agilent) on a high sensitivity DNA chip, with classification of a positive result based on fragment sizes ±5 bp of those expected, and peak concentration ≥500 pg/ul. In addition to the above controls, extraction and PCR negative controls were included, which substituted input genomic DNA for purified water.

### Statistics

Species richness and the evenness of their abundance were quantified using the Shannon–Weaver index ecological measure, calculated within MEGAN. Visualisation of samples was performed by hierarchical clustering using the UPGMA method and principal coordinates analysis (PCoA), all based on a matrix of Bray–Curtis distances calculated within MEGAN. Correlations and *t*-tests were performed within R (v 3.2.5) (*R Developement Core Team, 2015*).

**Table 1  Metagenomic study dataset.** Sequencing results for the eleven preterm samples sequenced and four replicates.

| Sample | Instrument | Read length | Raw PE reads | Surviving PE reads[*] | Surviving PE reads %[*] | Mean read length (bp) | Yield (Gbps) |
|---|---|---|---|---|---|---|---|
| Q19 | NextSeq | 151 | 10,765,181 | 9,748,620 | 90.6 | 118.5 | 2.3 |
| Q26 | NextSeq | 151 | 10,515,261 | 9,675,994 | 92.0 | 125.3 | 2.4 |
| Q83 | NextSeq | 151 | 10,272,541 | 9,434,574 | 91.8 | 128.1 | 2.4 |
| Q87 | NextSeq | 151 | 9,771,928 | 9,031,166 | 92.4 | 126.3 | 2.3 |
| Q89 | NextSeq | 151 | 10,573,383 | 9,686,447 | 91.6 | 130.5 | 2.5 |
| Q89 (r) | NextSeq | 151 | 10,746,440 | 9,824,221 | 91.4 | 129.1 | 2.5 |
| Q117 | NextSeq | 151 | 9,718,743 | 8,943,195 | 92.0 | 124.4 | 2.2 |
| Q142 | NextSeq | 151 | 9,847,000 | 9,059,108 | 92.0 | 129.0 | 2.3 |
| Q175 | NextSeq | 151 | 11,442,761 | 10,404,737 | 90.9 | 121.1 | 2.5 |
| Q189 | NextSeq | 151 | 9,385,745 | 8,667,469 | 92.3 | 125.2 | 2.2 |
| Q216 | NextSeq | 151 | 11,842,425 | 10,963,555 | 92.6 | 125.8 | 2.8 |
| Q219 | NextSeq | 151 | 5,761,388 | 5,295,708 | 91.9 | 118.4 | 1.3 |
| Q87 (r) | MiSeq | 301 | 8,061,151 | 5,710,557 | 70.8 | 193.0 | 1.1 |
| Q142 (r) | MiSeq | 301 | 4,101,014 | 3,234,121 | 78.9 | 204.1 | 0.7 |
| Q216 (r) | MiSeq | 301 | 12,778,363 | 10,848,709 | 84.9 | 224.0 | 2.4 |
| Mean | – | – | 9,705,555 | 8,701,879 | – | – | 2.1 |
| Total | – | – | 145,583,324 | 130,528,181 | 89.7 | – | – |

Notes.
[*] Three MiSeq replicate samples paired reads were merged during QC, therefore read number represent single reads.
(r) signifies replicate samples.

## RESULTS

### The healthy preterm metagenome

Using shotgun metagenomic sequencing we have captured an early snapshot of the antimicrobial resistance landscape within the gut microbiota of eleven premature infants who did not have proven sepsis or necrotizing enterocolitis. Infants were born either vaginally ($N = 6$) or by caesarean section ($N = 5$), with gestational ages ranging 24–31 weeks (mean 26.9 weeks). Ages of the infants at which the samples were taken ranged from 5 to 43 days (mean 25.7 days) (Table S1). A mixture of benchtop to medium throughput Illumina platforms were used to generate a dataset of 145.6 million paired end (PE) reads (51.4 Gbp sequence data) (Table 1), enabling us to characterise taxonomic and antimicrobial resistance profiles.

The eleven sequenced samples and four replicates were analysed using a blastx type analysis with filtering by the Lowest Common Ancestor (*Huson et al., 2007*; *Buchfink, Xie & Huson, 2014*), which enabled assignment of taxonomic labels for 71.5% of the reads within the complete dataset to at least the level of Kingdom (Table S2). As an alternative method, we also profiled the dataset using a marker based approach (*Truong et al., 2015*), which was highly congruent to species level relative abundances, as well as higher taxonomic ranks, to the blast based method used (Pearson $R = 0.9$–1.0) (Table S3). Replicate sequencing of samples also demonstrated reproducibility of the method, either by cluster analysis (Fig. 1) or pairwise correlations (Fig. S1).

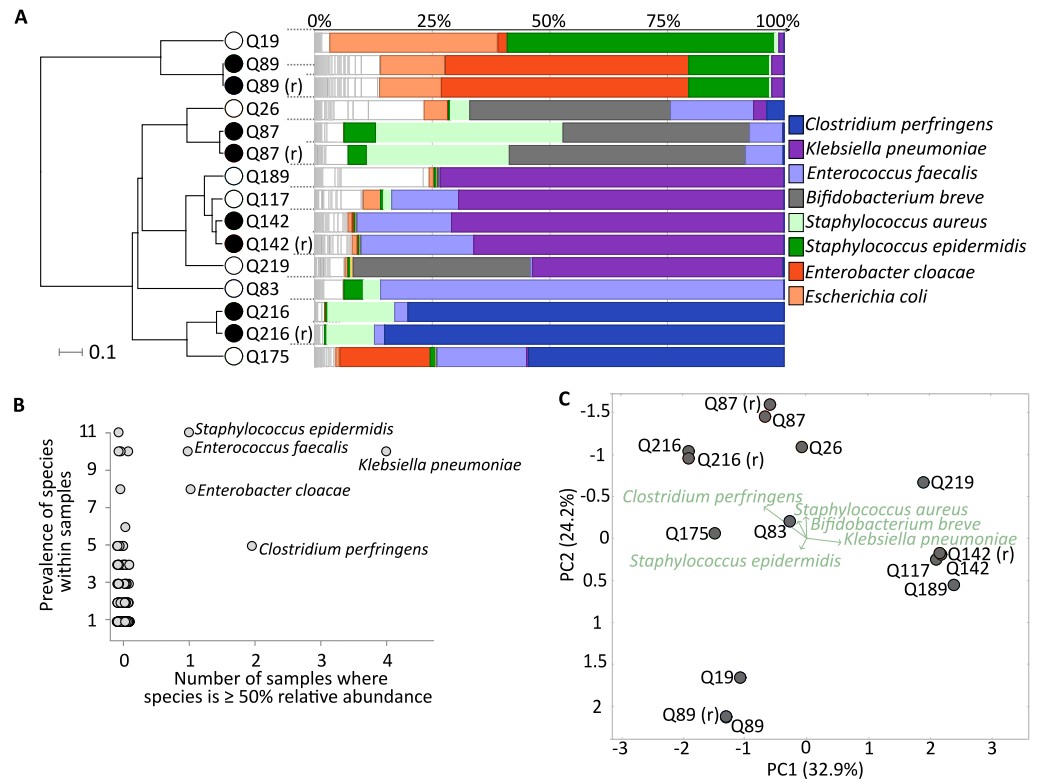

**Figure 1** **Healthy premature infant gut microbiome.** (A) Metagenomic profiles for the eleven preterm samples and four replicates at the species level. Samples clustered by UPGMA using Bray–Curtis distances shown on left, with replicates highlighted by filled nodes. Relative abundances by rank order shown on right, with the top 8 most abundant species coloured and labelled, leaving remaining species in white. (B) Dominant species, based on ≥50% abundance, shown on $x$-axis, with overall prevalence of the species across samples shown on $y$-axis. Sample number reflects eleven neonates as replicates are averaged. The five labelled species are present in five or more samples with at least one in >50% abundance. (C) Principal coordinates analysis (PCoA) using Bray–Curtis distances at the species level for all fifteen samples. Separation of the three broad sample groups shown by biplot of the top five species.

Moving to taxonomic composition, each sample was marked by a few highly abundant species, such as sample Q216 with 85.1% *Clostridium perfringens*, Q189 with 73.1% *Klebsiella pneumoniae*, and Q83 with 85.9% *Enterococcus faecalis* (Fig. 1A). In terms of prevalence, the previous three species, as well as *Enterobacter cloacae* and *Staphylococcus epidermidis*, were found at over 50% relative abundance in one or more samples. Furthermore, S. *epidermidis* and S. *aureus* were ubiquitous, ranging from 0.06% to 57.1% abundance in all samples (Fig. 1B). Principal coordinate analysis (PCoA) demonstrated three loose sample groups based on a high abundance of S. *epidermidis*, K. *pneumoniae*, and either B. *breve*, S. *aureus* or C. *perfringens* (Fig. 1C). In total we identified a non-redundant set of 172 species across all samples (see Table S4 for complete dataset).

## Prevalence of antimicrobial resistance
Before focusing on individual AMR genes, we measured the $\alpha$-diversity (Shannon–Weaver index) within each sample and, although a small sample set, compared this to

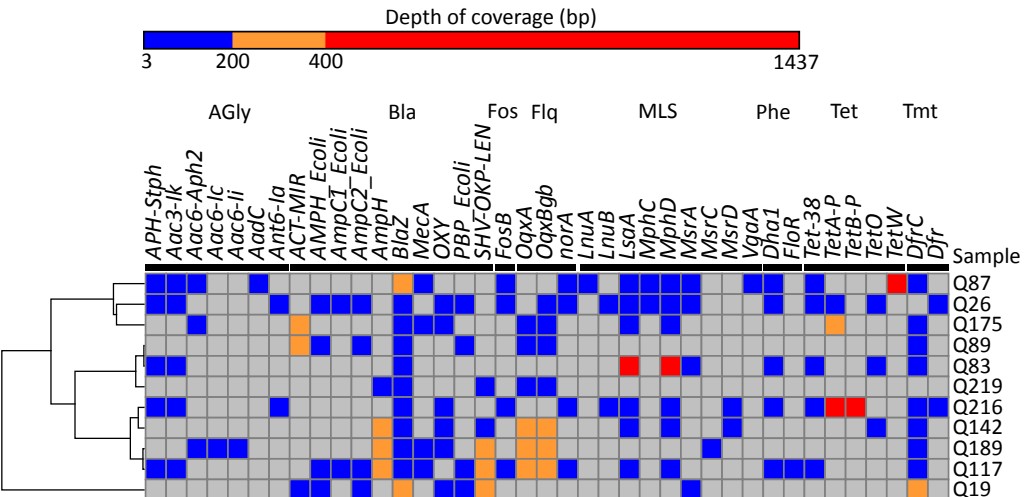

**Figure 2 Antibiotic resistance genes detected.** Heatmap showing distribution of the 39 AMR genes detected within the eleven metagenomic samples. Genes grouped by antibiotic class are demarcated by black lines under gene names: AGly (aminoglycosides), Bla ($\beta$-lactam), Fos (Fosfomycin), Flq (fluoro-quinolones), MLS (macrolide-lincosamide- streptogramin), Phe (phenicols), Tet (tetracyclines), Tmt (trimethoprim). Colours show read depth in bp: undetected (grey), 3–199 bp (blue), 200–399 bp (yellow), 400–1,437 (red). Rows clustered by UPGMA method using Euclidean distances.

the known antibiotic exposure of the preterm infants (Fig. S2). Eight of the eleven infants had received a course of prophylactic antibiotic treatment consisting of co-amoxiclav (Table S1), whilst a second course was administered to four infants, consisting of combinations of co-amoxiclav, tazocin or vancomycin. In total, exposure ranged from 2 to 8 days of antibiotics before samples were taken, excluding infants Q87 and Q89 which received no antibiotics. Antibiotics were also administered maternally to three infants (Q26, Q117 & Q189), but this did not include the two above infants with no antibiotic treatment. Sample diversity ranged from 0.9 to 2.9 (SD ± 0.5), but when compared to cumulative antibiotic exposure expressed in days, no significant difference was found between the taxonomic diversity and amount of antibiotic exposure for untreated and treated infants (unpaired $t$-test, $P = 0.17$) (Fig. S2). However, it is important to stress that the small and heterogeneous nature of the sample set will have reduced the power to detect differences between antibiotic exposure in this study, and so prevented any meaningful stratification by other clinical variables such as mode of delivery or day of life.

A mapping based approach against a comprehensive collection of acquired antibiotic resistance genes was next used to quantify AMR within the eleven metagenomes (*Inouye et al., 2014*). In total 143 AMR genes were identified, consisting of a non-redundant set of 39 different AMR genes (Fig. 2 and Table S5). Per infant, the average number of genes identified was 13 (ranging 5–22 genes), and AMR genes were found across eight different antibiotic classes, including aminoglycosides and fluoroquinolones (Table S6). Mean sequence coverage across the sequence database was 99.0%, and sequence divergence ranged from no difference to 12.3% (Table S6). In total over 1,600 alleles were searched for, and notable AMR genes not detected included those involved in carbapenem and

**Table 2 Antibiotic classes identified.** Major antibiotic resistant classes of genes identified within the eleven samples by SRST2. Columns show antibiotic type and number of genes found within class.

| Antibiotic type | Number of identified genes within class |
| --- | --- |
| $\beta$-lactam (Bla) | 10 |
| Macrolide-lincosamide- streptogramin (MLS) | 9 |
| Aminoglycosides (AGly) | 7 |
| Tetracyclines (Tet) | 5 |
| Fluoroquinolones (Flq) | 3 |
| Phenicols (Phe) | 2 |
| Trimethoprim (Tmt) | 2 |
| Fosfomycin (Fcyn) | 1 |

vancomycin resistance, the latter of which was administered to three preterm infants prior to sample collection (Table S1). The class most frequently detected were $\beta$-lactamases, comprising ten different genes (Table 2), of which the *blaZ* gene was present in every infant. Interestingly, within this set of $\beta$-lactamase genes, *mecA* was found in four infants (Q87, Q117, Q175, and Q189), and at a mean depth of coverage ranging from 3.9 to 52.2 bp (Fig. 2). *mecA* confers resistance to methicillin as well as other $\beta$-lactam antibiotics, and is carried on the SCCmec mobile element found across several *Staphylococci* species. Identification of four infants with potential methicillin resistant *S. aureus* (MRSA) or *S. epidermidis* (MRSE) carriage, along with high abundances and prevalence of both *S. aureus* and *S. epidermidis* species across the dataset, could indicate a significant reservoir for AMR transfer between the species, as well as highlight the seeding of the infant gut microbiome from an early stage.

## Focus on *S. aureus* species detected

Next, we wanted to understand the relationship of the *S. aureus* species within the *mecA* positive as well as negative samples, as the premature infants overlapped in time and so could harbour closely related strains. This was undertaken to firstly confirm *in silico* prediction of *mecA* using an established molecular based typing method, but also to push the metagenomic analysis further on what was known to be a challenging dataset owing to the range of identified *S. aureus* as described above, with relative abundance ranging from 0.06% to 39.8% (Table S4). We first tested the computational prediction of *mecA* experimentally using a multiplexed PCR typing method (*Milheiriço, Oliveira & De Lencastre, 2007*), which provides detection of the *mecA* gene, in addition to typing of the mobile element carrying the gene (SCCmec), although this component of the assay was beyond the scope of this study. Using this method we detected *mecA* presence correctly within the control strains, a methicillin resistant (MRSA) and susceptible (MSSA) strain (see 'Materials & Methods'), and three out of the four predicted *mecA* positive samples generated a positive *mecA* result (Table S7). The exception, sample Q87, generated the expected *mecA* amplicon size but the concentration of this fell below the threshold for detection (<500 pg/ul) and so was excluded.

In an attempt to understand strain relatedness directly from the metagenomic data, we undertook *in silico* MLST analysis using an *S. aureus* schema as well as metagenome assembly. The MLST was able to classify four of the eleven samples, all with different ST types—ST8, ST1027, ST22, ST25, although the last two had some degree of uncertainty in their assignment (Table S8). This suggests that for at least these four samples, the *S. aureus* strains are unrelated and unlikely a result of transmission. We were interested to know if *de novo* assembly of the metagenome could be utilised to resolve these and any of the remaining unclassifiable samples further. Following assembly and identification of *S. aureus* contigs (see 'Materials & Methods'), we found that it was not possible to capture more than a fifth of the expected genome size for the above unclassified samples, with an abundance of >3% necessary to achieve over 90% estimated capture, which was achieved in four cases (Table S9). Phylogenetic reconstruction of these four genomes alongside a collection of published *S. aureus* genomes (Table S10), provided confirmation of the diversity of *S. aureus* identified (Fig. S3), enabling placement across a global collection of strains.

## DISCUSSION

It is recognised that one of the most important public health threats worldwide is antimicrobial resistance. Here we report on the gut composition and AMR diversity for eleven healthy but premature infants. Recent studies have shown that the initial seeding of the infant gut microbiome is influenced by the microorganisms in the immediate environment, and whilst colonisation by bacteria with AMR genes has been demonstrated (*Brooks et al., 2014*), comparatively far fewer studies have investigated the gut microbiome of infants, fewer still preterm healthy infants. Interest has also increased on how the trajectory of the early gut microbiome is influenced to form the 'stable' adult microbiome. The preterm infant gut microbiome is very different compared to full-term infants (*Groer et al., 2014*), displaying a much lower diversity, particularly in anaerobes, with an increase in coagulase-negative *Staphylococci* and *Enterobacteriaceae* (*Adlerberth & Wold, 2009*); adult microbiomes are characterised by several hundred, mostly anaerobic bacterial species (*Adlerberth & Wold, 2009*). We found a similarly low level of species diversity across all metagenomes, with each sample dominated by a few highly abundant species, including *C. perfringens*, *K. pneumoniae* and members of the *Staphylococci* and *Enterobacter* genera. Presence of such species are in common with previous studies on the premature gut microbiome (*Groer et al., 2014*; *Gibson et al., 2016*).

Interestingly, each metagenome profile displayed a different dominant few species, clustering into three loose groupings. This could reflect the dynamic nature of the early establishing gut, with the preterm infant microbiome acquiring an increased diversity of bacteria and subjected to a great amount of change until it matures into what is recognised as a more 'stable' microbiome. Although we found no correlation between diversity and antibiotic exposure, with infants treated with either no antibiotics (including during pregnancy), to up to 8 days of antibiotic administration, effects such as relatively small sample size, as well as day of life of sample and normal gut development are biases to this finding, which is contrary to other studies within infants (*Greenwood et al., 2014*;

*Merker et al., 2015*). It could be that at this very early stage, the microbiota is influenced to a greater extent by seeding during birth from the mother and environment than antibiotic treatment, or that not enough time has passed to detect differences from the antibiotics administered; larger sample numbers would be required, alongside longitudinal studies and parallel maternal sampling to better understand the development of diversity.

A threat to this development is the acquisition of antibiotic resistant bacteria, which can potentially seed the infant microbiome. Coupled with the high rate of horizontal gene transfer within the commensal community (*Stecher et al., 2012*), the preterm infant gut microbiome has the potential to be a reservoir for AMR. With dominance of the preterm gut by species known to carry clinically relevant antibiotic resistance, we next quantified the burden of antibiotic resistance genes within the infant's faecal flora, which identified an average of 13 genes per infant. Previous targeted or functional studies based on infants have found some of the AMR genes also identified here, including those for Tetracycline (*tet*) (*Gueimonde, Salminen & Isolauri, 2006*; *Alicea-Serrano et al., 2013*) and $\beta$-lactam (bla) (*Fouhy et al., 2014*). In a wider context, it is known that AMR genes are a common feature of bacterial populations, found in communities inhabiting the soil, rivers and even deep-sea sediment (*Knapp et al., 2010*; *Qin et al., 2011*; *Kittinger et al., 2016*). Therefore, whilst their presence in the human gut microbiome should be of little surprise (*Bailey et al., 2010*), identification of genes such as *mecA* demonstrates the prevalence of some clinically significant resistant bacteria from birth.

One of the advantages of the method used in this study is the utility of the results generated, enabling multiple avenues of questions to be addressed. However, short read sequencing remains a challenge when applied to the linkage of resistance elements, such as *mecA*, to specific genome sequences (strains), which is made difficult by the nature of metagenomic samples containing multiple alleles from different closely related species, as well as potentially multiple strains of the same species. Secondly, the methods used here were inherently restricted to identification of known AMR genes found within the ARGannot database used in this study, which contains those genes involved in acquired resistance only, therefore chromosomal mutations, such as those conferring resistance to rifampicin as well as novel resistance genes would have been missed, leading to potential underrepresentation of resistance in this study.

## CONCLUSIONS

The healthy preterm infants sampled within this study harboured multiple AMR genes, representing a potential reservoir for later disease onset. In particular, detection of clinically important AMR genes, such as *mecA*, highlights the need to further understand the impact that this reservoir could have on later treatment regimes. From a methodology point, this approach was able to provide a comprehensive snapshot of the complete taxonomic diversity and resistome in one assay. Although tracking of the movement of such AMR genetic elements would be enhanced by improved handling of the dynamic ranges of abundances; different methods at the level of sample preparation, such as sample normalisation, may offer potential answers to such hurdles. Overall this study leads to

questions such as how this resistance potential contributes to later clinical intervention or disease onset, and if antibiotic treatment without knowledge of prior AMR burden could lead to unintentional harm. More broadly, this and other studies show the great promise that shotgun metagenomics holds for clinical microbiology.

### Abbreviations

| | |
|---|---|
| **AMR** | antimicrobial resistance |
| **GI** | gastrointestinal |
| **PCR** | polymerase chain reaction |
| **qPCR** | quantitative polymerase chain reaction |
| *BLAST* | basic local alignment search tool |
| **SCCmec** | staphylococcal chromosome cassette *mec* |
| **LCA** | lowest common ancestor |
| **MLST** | multilocus sequence typing |
| **MALDI-TOF** | matrix-assisted laser desorption/ionization –time of flight |
| **PCoA** | principal coordinates analysis. |

### Funding

This study was supported by a programme grant (to J.Simon Kroll) from The Winnicott Foundation and generous additional funding from Micropathology Ltd., Meningitis Now and the National Institute for Health Research (NIHR) Biomedical Research Centre based at Imperial Healthcare NHS Trust and Imperial College London. The funders had no role in study design, data collection and analysis, decision to publish, or preparation of the manuscript.

### Grant Disclosures

The following grant information was disclosed by the authors:
The Winnicott Foundation.
Micropathology Ltd.
National Institute for Health Research (NIHR) Biomedical Research Centre.
Imperial Healthcare NHS Trust and Imperial College London.

### Competing Interests

The authors declare there are no competing interests.

### Author Contributions

- Graham Rose conceived and designed the experiments, analyzed the data, contributed reagents/materials/analysis tools, wrote the paper, prepared figures and/or tables, reviewed drafts of the paper.
- Alexander G. Shaw conceived and designed the experiments, performed the experiments, analyzed the data, wrote the paper, reviewed drafts of the paper.
- Kathleen Sim conceived and designed the experiments, performed the experiments, reviewed drafts of the paper.

- David J. Wooldridge and Ming-Shi Li performed the experiments, contributed reagents/materials/analysis tools, reviewed drafts of the paper.
- Saheer Gharbia reviewed drafts of the paper.
- Raju Misra and John Simon Kroll conceived and designed the experiments, reviewed drafts of the paper.

### Human Ethics

The following information was supplied relating to ethical approvals (i.e., approving body and any reference numbers):

The study was approved by West London Research Ethics Committee (REC) Two, United Kingdom, under the REC approval reference number 10/H0711/39. Parents gave written informed consent for their infant to participate in the study.

### DNA Deposition

The following information was supplied regarding the deposition of DNA sequences:

Database: EBI European Nucleotide Archive.

Accession number: PRJEB15257.

### Data Availability

The raw data has been supplied as a Supplementary File.

### Supplemental Information

Supplemental information for this article can be found online at http://dx.doi.org/10.7717/peerj.2928#supplemental-information.

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
