# Peer review of "Antibiotic resistance potential of the healthy preterm infant gut microbiome"

_PeerJ, doi:10.7717/peerj.2928_

## Round 0.1 · original submission · Major Revisions

· Academic Editor

Major Revisions

Dear Dr. Kroll, Dear Dr. Rose,

Two reviewers have seen and reviewed your fine manuscript. Although both think that it merits publicqtion at some point, they have raised some issues that have to be addressed nbeforwe a further decision can be taken. I support the reviewers points, so please be extra careful to not overstate your findings due to the low sample size.

Please provide a point-by-point response letter to the reviewers' comments.

Many thanks again for submitting your work to PeerJ.

Best regards,
Christine Josenhans

Reviewer 1 ·

Basic reporting

The authors have done a fine job in the writing, background and scholarly references to the literature.

I picked up a few errors/typos though:

Line 202, change "Presence absence" to "Presence/absence".

Line 253, change to "...were used to analyse correlations...".

Line 330, change "stains" to "strains".

Line 344, clarify what is meant by "But when possible".

Line 402, "hypotheses" should be generated or tested, not "asked". An alternative would be "multiple avenues of questions to be addressed".

Line 457-458, the title in this ref is written twice.

Experimental design

The technical design regarding the sampling handling, sequencing and data processing and handling are all sound. It would be further nice if the authors could offer a bit more explanation regarding the technical reasons for needing to do additional SCCmec typing separate from the pure metagenomic approach.

A rather big weakness in the design, however, is the very low sample size which I feel precludes making any statements regarding the effects of antibiotic use or mode of delivery. I presume, however, that this mostly reflects the difficulties in obtaining a higher number of healthy pre-term samples, and the study nonetheless provides a number of observations important to the community. The issue with sample size should thus be more carefully considered when stating the conclusions (see below). It may also help to note the frequency with which one can obtain healthy preterm infants with no antibiotic use.

Validity of the findings

As mentioned by the authors themselves in the Discussion (lines 379-386), the low sample size and heterogeneity of age and antibiotic administration is a major weakness. It is hard for me to imagine that the statistical models used by the authors would have any power to detect influences of antibiotic use or mode of delivery in a human setting with these numbers. Thus, I would recommend the authors to be more up front with this, especially in order to prevent the propagation of potentially misleading false negative results.

Additional comments

Please emphasise what you CAN say, and be more clear about what you cannot say.

Reviewer 2 ·

Basic reporting

No comments

Experimental design

The research question was concise and relevant. The results are indeed novel and represent a significant contribution to the field. Most experiments were adequately performed and can be reproduced elsewhere.

The methods used to estimate diversity indexes is missing in the Materials and Methods section.

Validity of the findings

Data is robust and controlled. The conclusions are well stated and supported.

Additional comments

Since the authors do not present % of coverage and identity for the genes that they detected, it is hard to say whether they are dealing with new alleles. I recommend addressing this issue. In line with this comment, the authors should discuss that the bioinformatic approach followed likely overlooks genes not included in the ARGannot database.

It would be advisable to sequence the SSCmec elements as novel and known elements can have equal sizes.

All data related to the similarity of the study and global S. aureus genomes should be removed from the manuscript. This information has nothing to do with the research question and is not required to understand the data.

This reviewer believes that the manuscript and the interpretations that can be derived from it will improve if the authors explicitly present the molecular context of the resistance genes identified in the screening. Were they included into mobile genetic elements? of what kind? Transposons? Plasmids?, of what families?.

It would also be important to know the relative abundance of the bacterial genera, resistance genes, and mobile genetic elements in the gut microbiota of the babies“ mothers.

---

## Round 0.2 · accepted · Accept

· Academic Editor

Accept

Dear Dr. Rose, dear Dr. Kroll,

Thank you for improving your revised manuscript according to the valuable suggestions of our reviewers.

Congratulations on your fine work.

Kind regards,

Christine Josenhans
Academic Editor of PeerJ

Reviewer 1 ·

Basic reporting

no comment

Experimental design

no comment

Validity of the findings

no comment

Additional comments

The authors have sufficiently addressed the issues I raised with the previous version of the manuscript, nice work.